# Automated Particle Tracing & Sensitivity Analysis for Residence Time in a Saturated Subsurface Media

**Md Abdullah Al Mehedi** [1,*] and **Munshi Md. Shafwat Yazdan** [2]

1    Civil & Environmental Engineering, Villanova University, Villanova, PA 19085, USA
2    Civil & Environmental Engineering, Idaho State University, Pocatello, ID 83209, USA; yazdmuns@isu.edu
*    Correspondence: mmehedi@villanova.edu; Tel.: +1-(267)-721-0317

**Abstract:** Residence time of water flow is an important factor in subsurface media to determine the fate of environmental toxins and the metabolic rates in the ecotone between the surface stream and groundwater. Both numerical and lab-based experimentation can be used to estimate the residence time. However, due to high variability in material composition in subsurface media, a pragmatic model set up in the laboratory to trace particles is strenuous. Nevertheless, the selection and inclusion of input parameters, execution of the simulation, and generation of results as well as post-processing of the outcomes of a simulation take a considerable amount of time. To address these challenges, an automated particle tracing method is developed where the numerical model, i.e., flow and reactive transport code, MIN3P, and MATLAB code for tracing particles in saturated porous media, is used. A rectangular model domain is set up considering a fully saturated subsurface media under steady-state conditions in MIN3P. Streamlines and residence times of the particles are computed with a variety of seeding locations covering the whole model surface. Sensitivity analysis for residence time is performed over the varying spatial discretization and computational time steps. Moreover, a comparative study of the outcomes with Paraview is undertaken to validate the automated model ($R^2$ = 0.997). The outcome of the automated process illustrates that the computed residence times are highly dependent on the accuracy of the integration method, the value of the computational time step, $\Delta t$, spatial discretization, stopping criterion for the integration process of streamlines, location, and amount of seed points. The automated process can be highly beneficial in obtaining insights into subsurface flow dynamics with high variability in the model setup instead of laboratory-based experimentation in a computationally efficient manner.

**Keywords:** MIN3P; MATLAB; particle tracing; streamlines; subsurface flow; numerical discretization





## 1. Introduction

The capability of numerical simulation of flow as a tool for investigating science in general and hydrogeologic sciences, in particular, has reached a new level in recent days [1,2]. The direction of this type of research represents a promising future aimed at providing useful alternatives to laboratory experiments where realistic input parameters into hydrogeologic simulations are strenuous [3–6]. Water flow mechanisms in the subsurface media are subject to many fields [7–10]. Some of them are applications in the soil sciences and subsurface hydrology, e.g., water flow path in the subsurface through the infiltration process [11,12]. The new development in high-performance computing, in terms of hardware and networking with sufficient bandwidth and low latency, as well as in terms of advanced algorithms capable of simulating complex problems in subsurface hydrology, makes it possible today to trace particles in porous media [13–15]. Massless particle tracing is a crucial advancement toward obtaining a better intuition of the complex phenomena that occur inside the subsurface system. This progress makes it feasible to analyze various factors that play a key role in the vadose zone hydrology as well as hyporheic zone, where important biogeochemical reactions of stream and groundwater solutes occur with a

crucial impact on nutrient cycling in the fluvial system [16,17]. These massless particles are advected throughout the computational domain, and they are widely used to sample relevant data and to visualize the streamlines through time and space [18]. An example of particular interest in subsurface simulations is the residence time, a quantity that is attached to particles involved in investigating the water flow [19]. It basically measures the time that particles remain in their traveling throughout a certain region of analysis and help to characterize stagnation regions or fast-flowing regions [20]. However, a robust velocity vector field of a model domain is required to simulate the particle tracks. In this study, the MIN3P code is used to solve Richard's equation [21]. A regular rectangular model domain under fully saturated steady-state conditions is considered for the study.

Hyporheic residence times and streamlining resulting from groundwater and surface water exchange are strongly influenced by stream bed configurations. An undulating streambed alters the currents, creating hydraulic gradients along with the soil-water interface that works as a driver of groundwater and surface water exchange [22,23]. The automated generation of streamlines in the subsurface domain, with various streambed setups and subsurface characteristics, could help to bring a better understanding of the process and behavior of the streamlines and residence time distribution under varying streambed conditions, whereas this cannot be efficiently performed in the case in the laboratory experiments [24]. Moreover, it is not possible for repeatability of results, a high number of variations, greater insight into the three-dimensional system, or understanding of individual streamlines or residence time distribution to be well executed in the lab-based experimentation. These circumstances create the need for automated numerical modeling to understand better the principal subsurface water flow mechanisms for research and work, which are arduous to isolate under a natural in-stream environment. Therefore, in line with this assertion, this paper introduced a novel automated framework to compute the subsurface water flow and corresponding residence times using MIN3P and MATLAB. The objectives of this paper are to track particles seeded from the surface of a fully saturated porous model domain, thus computing the residence time, and perform sensitivity analysis over the residence time for varying spatial discretization and computational time steps. MATLAB is used for automatically executing the MIN3P code generating the velocity vector field in the subsurface domain and importing it within its interface. MATLAB is also used for developing a code to trace particles seeded from the top surface of the model and generate streamlines. In addition, an inclusive comparative study of the performance of developed code in MATLAB and a 3D vector field visualization application, Paraview, was performed to validate the MATLAB code [22].

## 2. Materials and Methods

The basic equations for water flow in the saturated subsurface media provide

$$S\frac{\partial h}{\partial t} - \nabla.\boldsymbol{q} = Q \tag{1}$$

$$\boldsymbol{q} = -\boldsymbol{K}.\nabla h \tag{2}$$

The above equation is solved for the variables, hydraulic head, $h$, and the water flux $\boldsymbol{q}$, $S$, $\boldsymbol{K}$, and $Q$ denote the specific storage coefficient, tensor of hydraulic conductivity, and the general source/sink function, respectively [21,23]. The water flux, $\boldsymbol{q}$, is substituted by the Darcy equation to obtain the final Richards-type equation:

$$S\frac{\partial h}{\partial t} + \nabla.(\boldsymbol{K}.\nabla h) = Q_h + Q_{hw} \tag{3}$$

In the equation above, the source/sink term is denoted by $Q = Q_h + Q_{hw}$, suitably split into a supply term $Q_h$ and a well-type term $Q_{hw}$. The Richards equation shows the water movement in unsaturated soils. It is expressed by the nonlinear partial differential equation. The equation is based on Darcy's law for groundwater flow conception [24–28].

In this study, a fully saturated porous media is considered for analyzing particle tracks if seeded from the surface of the model domain. Particles go along the path, which is determined by the velocity vector field present in the model domain. The time elapsed by a particle from entering the porous media to getting out of the system is called residence time. The mean residence time (MRT) is the average residence time of multiple particles seeded from a zone of certain study areas. The mean residence time (MRT) is an important factor in the hydrogeological concepts where there is a mixing of shallow groundwater (vadose zone) and surface water [27,29].

Trajectories of the particle streamlines are computed in the automated particle tracing process to analyze the residence time distribution. MIN3P is a multicomponent reactive transport code for variably saturated porous media. MIN3P is used for generating the velocity vector field required for creating streamlined trajectories [30]. In the automated generation of the velocity vector field, a fully saturated porous media under steady-state conditions is considered. A regular rectangular model domain is set up in MIN3P code. Residence time from individual particles is calculated and recorded in an automated particle tracing process to further analyze the impact of seeding locations over the value and distribution of streamlines and thus residence time. In this study, the model domain is divided into four zones having gravel and sand as porous media. The typical grain size of gravel and sand range from 0.2 mm to 0.16 mm and 250 μm to 1–2 m, respectively. In MIN3P, physical characteristics of porous media and water flow parameters are characterized by porosity and hydraulic conductivity for the fully saturated model domain. Porosity was assumed to be 0.33 for both sand and gravel. On the other hand, the hydraulic conductivity of sand and gravel was assumed to be $3.3 \times 10^{-3}$ m/s and $8.3 \times 10^{-2}$ m/s, respectively.

## 2.1. Automated Generation of Velocity Vector Field

A base model is set up using the MIN3P code, which solves the Richard's equation to compute water flow through the subsurface medium. A fully saturated porous model domain in a steady state is used for the analysis. A 3D computational space is set up based on the dimensions of X:Y:Z = 1:0.1:0.1 m (Figure 1). Constant head boundaries (1st-order Dirichlet) condition is assumed by the linear drop in the hydraulic head. In this study, the 1st-order Dirichlet boundary condition is assumed to keep the hydraulic head boundary condition constant with no temporal variation. Different numbers of control volumes were considered along x, y, and z for spatial discretization. For particle tracking, the number of control volumes along x, y, and z was considered to be 450, 30, and 25, respectively. The unit of the time step is fixed as hours for this study.

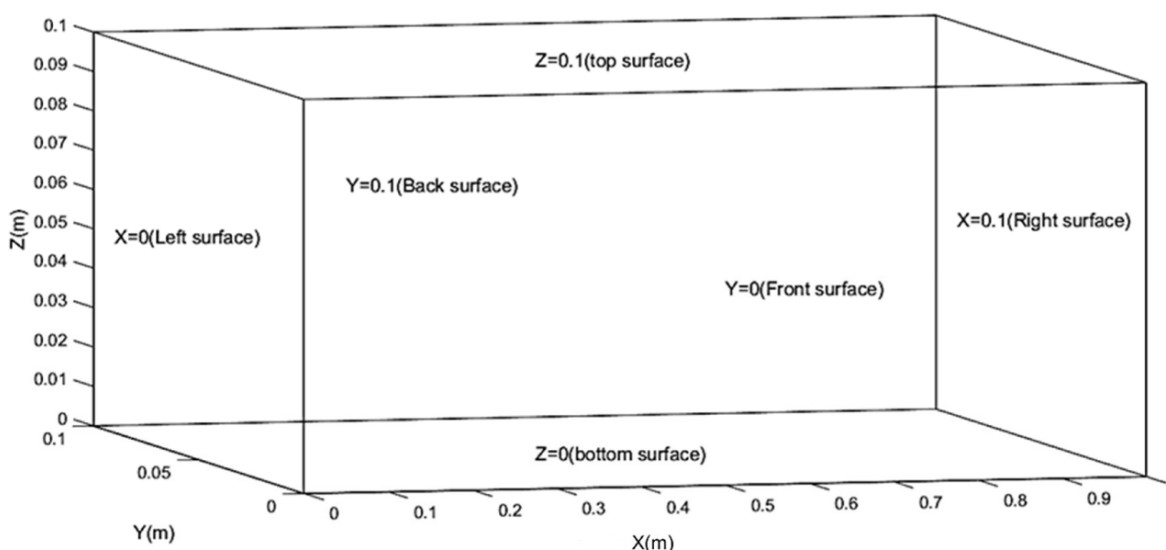

**Figure 1.** A 3D fully saturated porous model domain was set up based on the dimensions of X:Y:Z = 1:0.1:0.1 m.

The code includes an adaptive time-stepping scheme, which only requires the specification of a minimum and maximum computational time step. The minimum time step is used initially, and the time step will increase to maximize efficiency. Velocity components $(v_x, v_y, v_z)$ at the center of each computational cell, along with the coordinates $(x, y, z)$, are calculated by MIN3P. The initial condition is defined as fully saturated flow by the hydraulic head. The distribution of this parameter can be discretized across the model domain by means of zones. The initial condition for the zone is set to be 0.1 m. The initial condition is given in terms of the hydraulic head. In this study, four boundary zones, namely sand1, gravel1, sand2, and gravel2 in the z plane, are defined. The extent of the boundary zones is considered to be x = 0 to 0.25, y = 0 to 0.1 for sand1, x = 0.25 to 0.5, y = 0 to 0.1 for gravel1, x = 0.5 to 0.75, y = 0 to 0.1 for sand2 and x = 0.75 to 1, y = 0 to 0.1 for gravel2. The boundary conditions for sand (sand1 and sand2) and gravel (gravel1 and gravel2) zones are considered to be first-type (Dirichlet) boundary conditions with a value of 0.1 and 0.101 m (hydraulic head or pressure head), respectively.

## 2.2. Automated Particle Tracing Using MATLAB

Particles seeded from the surface of the model domain are traced in an automated process using MATLAB. The entire process of importing the subsurface flow field from MIN3P, applying the numerical method to compute particle flow path, and generating the distribution of streamlines and residence time is controlled in MATLAB. The fourth-order Runge–Kutta method is used in this study to trace hyporheic streamlines. The vectors considered in the subsurface water flow field can be written as follows

$$P_{i+1} = P_i + (1/6)\, \Delta t \vec{v}_i + (1/3)\, \Delta t \vec{v}^{\mathbf{1}}_{i+1} + (1/3)\, \Delta t \vec{v}^{\mathbf{2}}_{i+1} + (1/6)\, \Delta t \vec{v}^{\mathbf{3}}_{i+1} \qquad (4)$$

where $P_i$ denotes the beginning position of the streamline, $\Delta t$ is the time step, $\vec{v}^{\mathbf{1}}_{i+1}$ is the flow vector corresponding to the point $P_i + (\frac{1}{2})\Delta t \vec{v}_i$, $\vec{v}^{\mathbf{2}}_{i+1}$ is the flow vector corresponding to the point $P_i + (\frac{1}{2})\Delta t \vec{v}^{\mathbf{1}}_{i+1}$ and $\vec{v}^{\mathbf{3}}_{i+1}$ is the vector corresponding to the point $P_i + \Delta t \vec{v}^{\mathbf{2}}_{i+1}$.

There are six variables providing the coordinates of the grid points $(x, y, z)$ and the velocity $(v_x, v_y, v_z)$ in the velocity field file. The velocity field file generated from MIN3P is imported in MATLAB, keeping only the variables of locations and their corresponding velocities $(x, y, z$ and $v_x, v_y, v_z)$ and stored in a 2D array. After getting the particle's seeding location, the program searches the nearest location, or whether the particle stays exactly on any meshing point (computational nodes generated by MIN3P) and the corresponding velocity (stored in computational cell center), forward Euler's method is used for the numerical integration of streamlines. The program searches the nearest location of the seeded particle to predict the next location. Using the formula $P_1 = p + \Delta t \vec{v}$, where $p$, $\Delta t$, $\vec{v}$, and $P_1$ are the particle's current location, computational time step, velocity vector of the nearest point, and predicted location of the particle, respectively. Multiple particles showed in the can be seeded in a uniformly distributed manner over the surface of the model. The integration stop criteria are defined in such a way that when a particle reaches any boundary plane of the computational space (x = 0, x = 1, y = 0, y = 0.1, z = 0 and z = 0.1), it stops. The full workflow of the automation process is shown in Figure 2. In the first step, model geometry with spatial and temporal discretization, initial, and boundary conditions are set for the model run. After obtaining the results in the form of velocity vectors from MIN3P, MATLAB code is used to preprocess the vector data. The particle is traced by generating the streamlines. The residence time is taken by the particle estimated in MATLAB. Finally, the streamline maps and residence time distribution are achieved from the automated process.

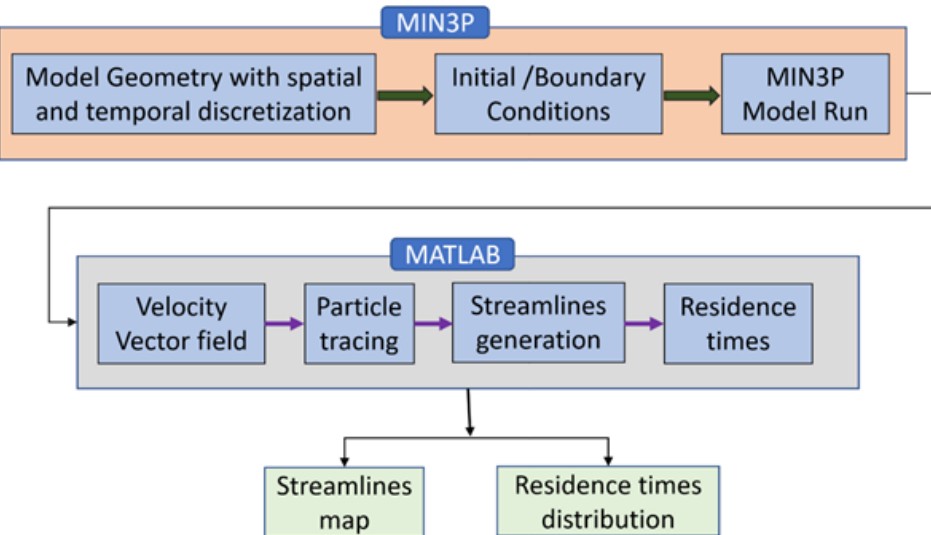

**Figure 2.** Workflow of the automation process to generate the streamlines.

*2.3. Comparison with Paraview*

Paraview offers 3D visualization for the fluid flow field. Researchers and scientists from many disciplines use it to analyze and visualize scientific data [31]. Flowfields can be investigated using glyphs to the points. Water streamlines can be generated using a constant computational step and/or adaptive integrators. Particle paths can be extracted from temporal data sets [32,33]. A comparative study is performed between the automated process in MIN3P and MATLAB and Paraview to observe and quantify the variation of the outcomes, i.e., streamlines and residence time distribution.

*2.4. Sensitivity Analysis of Residence Time*

Sensitivity analysis due to the variation of the spatial and temporal discretization is performed to observe the response of the residence time. In MIN3P, the dimensions of the simulation are specified (3D), and the geometry of the domain is defined. The spatial discretization of the model is based on a control volume method in MIN3P, and the domain is regular. Within each of the computational zones, the grid spacing will be uniform; however, the spacing may differ between zones. The accuracy of computing streamlines largely depends on the computational time step ($\Delta t$). The numerical integration technique applied for computing particle's path line and thus the residence time is based on the forward Euler method. In this method, particle positions are predicted by simply adding the particle's current position and the product of the computational time step and interpolated velocity at the particle's current position. In mathematical expression, it can be shown as $P_{i+1} = P_i + \Delta t \vec{v}_i$, where $P_{i+1}$ is the particle's predicted position, $P_i$ is the particle's current position and $\Delta t \vec{v}_i$ is the product of the computational time step and interpolated velocity at the particle's current position.

## 3. Results

*3.1. Distribution of Streamlines and Residence Time*

Particles were traced by seeding from the surface of the model domain. Z coordinate was assumed to be fixed as the particle seeding is feasible from the surface of the model domain, and that is the maximum z plane. In Figure 3, the distribution of streamlines generated from the automated process in MIN3P, and MATLAB is shown. The red asterisks denote the seeding locations. A representative y-plane is considered as no velocity variation is assumed in the direction of the y-axis. The depth of the traced streamlines can be clearly seen in Figure 3.

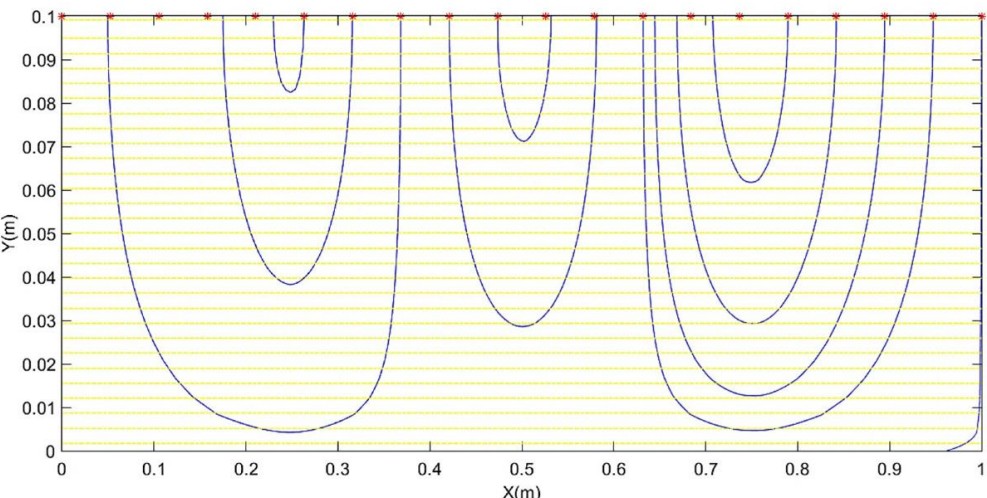

**Figure 3.** Streamline distribution at y = 0.05 plane show variations within x and z horizon.

In Figure 4, the distribution of the residence times for multiple y-planes (multiple colors in the legend) is shown. The range of the residence times found is 0–1.43 h depending on the seeding locations. Maximum residence time, 1.43 h, was found at the maximum extent in the x direction, i.e., x = 1 plane. Lengths of the streamlines thus the residence times are greater in the region of x = 0.75 to x = 1 with y = 0.01 to 0.09. On the other hand, streamlines also existed in the region of x = 0.25 to x = 0.5 and y = 0.01 to 0.09, where comparatively lower residence times were observed.

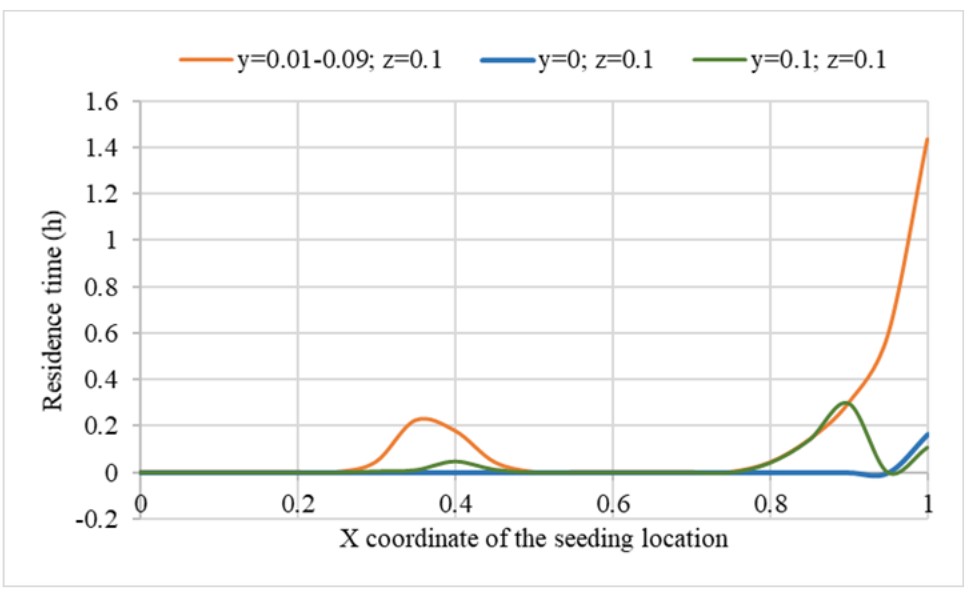

**Figure 4.** Residence time with varying x and y coordinates of the seeding locations.

As the velocity component in the y direction is negligible (maximum is $1.5543 \times 10^{-5}$ m/h), only a representative y-plane (y = 0.05) was selected to reduce the computational effort so that the number of streamlines could be increased. With only 21 seeding locations along the x direction on the surface, it is not feasible to apply any statistical analysis. Keeping this situation in mind, 2D (x and z) analysis was performed with a greater amount of seeding points such as from 50 to 2000 seeding points from the surface were applied to obtain a representative mean residence time. However, some zones are continuous upwelling zones, where no streamline was found. Those zones are x = 0 to x = 0.25 and x = 0.25 to x = 0.75. Other locations in the model domain contribute streamlines.

In Figure 5, it can be observed that with the increase in no. of particles seeded from the model surface, residence time increases. However, a stable condition was reached by increasing the no. of seeding locations. Up to 2000 seeding locations were chosen for analyzing the residence time. For instance, maximum residence times for particles seeded in the region x = 0.25 to x = 0.5, y = 0.05, z = 0.1, was recorded 0.22 h when the no. of seeding location was set to be 21. On the other hand, when the no. of seeding location was 1000, this value of maximum residence times increased to 1.12 h. Similarly, maximum residence times for particles seeded in the region x = 0.75 to x = 1, y = 0.05, z = 0.1, was recorded 1.435 h and 1.634 h when the no. of seeding location were 21 and 1000. However, when the number of seeding locations increased further up to 2000, a negligible change in the residence times was observed. The maximum residence times were observed at 1.22 h and 1.63 h when the number of seeding locations was 2000 for these two zones mentioned above.

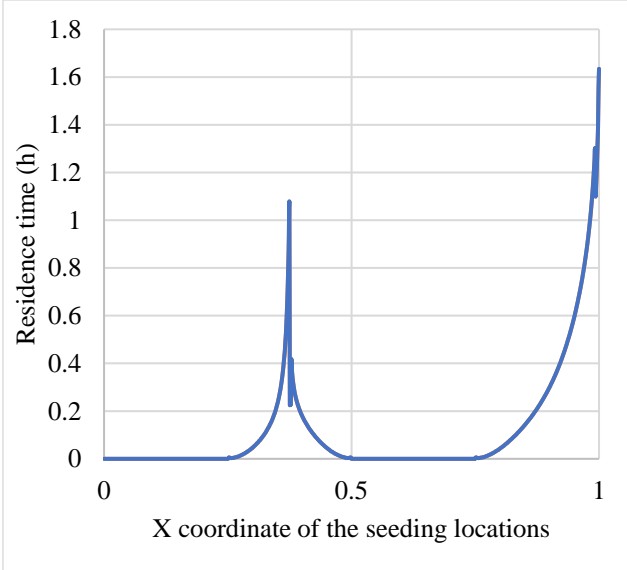 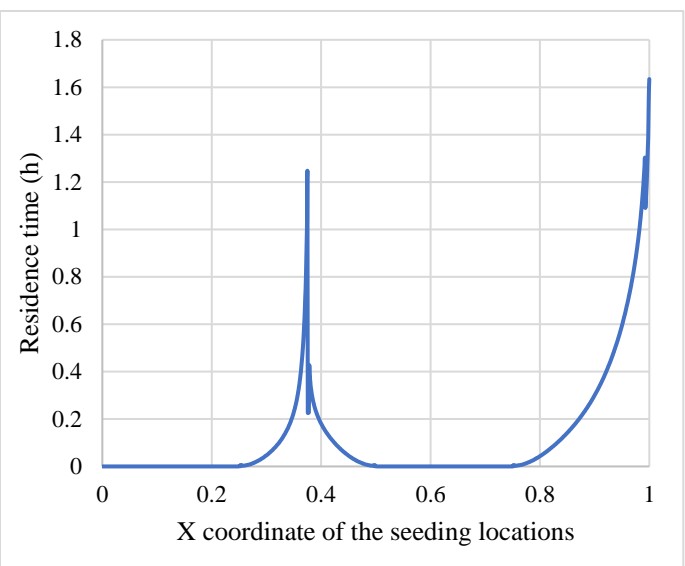

**Figure 5.** Variation of residence time with varying x coordinates (y = 0.05, z = 0.1) of the seeding locations. Number of seeding points are 1000 and 2000.

*3.2. Mean Residence Time*

A decreasing trend is observed while considering the mean residence time with an increasing number of particles seeded from the surface. Mean residence time is measured over the whole x horizon (x = 0 to x = 1) at y = 0.05 and z = 0.1. Zone from x = 0.25 to x = 0.5 and x = 0.75 to x = 1 considered as the selected streamlines and thus approximately 50% of the whole model domain constitute the streamline. The rest of the model domains do not constitute any streamlines, and because of that, their residence time is 0. Approximately 50% area of the model domain did not contribute any residence time and was considered when calculating the mean value for the whole x horizon (x = 0 to 1). The calculated mean residence time in such a manner is not representative of the whole model domain as the mean value is calculated from the data of 50% of the x horizon. However, the mean residence time reached 6.924 h from 7.787 h when the seeding location increased from 50 to 2000 and obtained a stable condition where the mean residence time did not change considerably with the increase in seeding location illustrated in Figure 6.

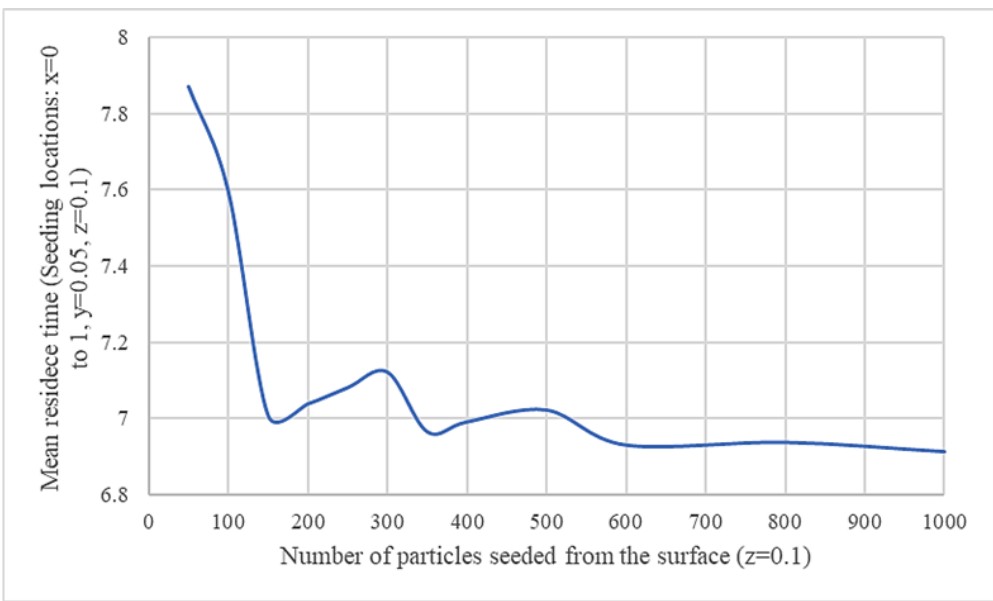

**Figure 6.** Variation of the mean residence time with the change in number of particles seeded from the model surface.

*3.3. Comparison with Paraview*

A comparative study is performed with Paraview to validate the outcomes of the developed code shown in Figure 7. Residence times obtained from both Paraview and MATLAB code were plotted to perform regression analysis. It is evident that the difference in the results is negligible. It is also plotted to measure the trend line and $R^2$. Approximately, every data point is on the trend line, and the value of R-squared is found to be 0.99986, which indicates the difference between measured residence time from Paraview and the code is negligible. This analysis was not performed for other y-planes because there is no change in streamlines and residence times along the y horizon due to negligible velocity components. Only the z = 0.09 plane is considered for the analysis.

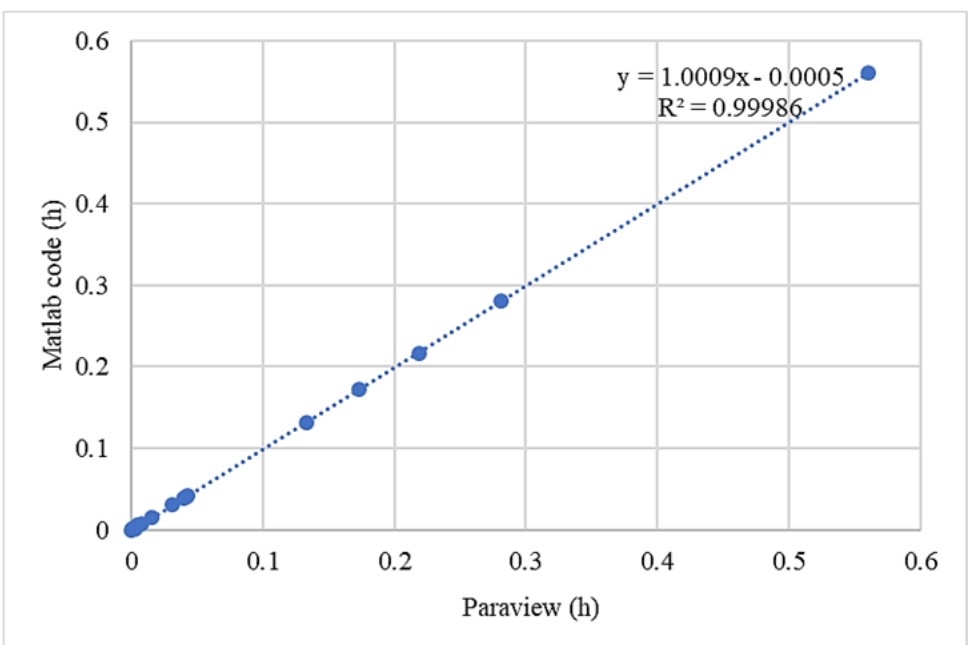

**Figure 7.** Comparison of residence times obtained from Paraview and MATLAB code.

Another comparative study is also performed while considering computational time step (dt) ranges from 0.0001 h to 0.02 h (0.36 s to 1 min 12 s) to observe the change in residence times for the same seeding locations (x = 0 to 1, y = 0.05 and z = 0.09) showed in Figure 8. As the computation time step increased, the difference between measured residence time from Paraview and the code was also increased. This is because the error in numerical integration from the forward Euler method increases with the increase in the computational time step in MATLAB code. On the other hand, Paraview uses a fixed time step or an adaptive time step where the error is minimized. In Figure 8, it can be seen that when the value of dt is smaller, Paraview and MATLAB code generate approximately the same results. However, when the value of dt is increased to a higher value, the accuracy in numerical integration for computing streamlines in MATLAB code reduces. As a result, the value of $R^2$ decreases.

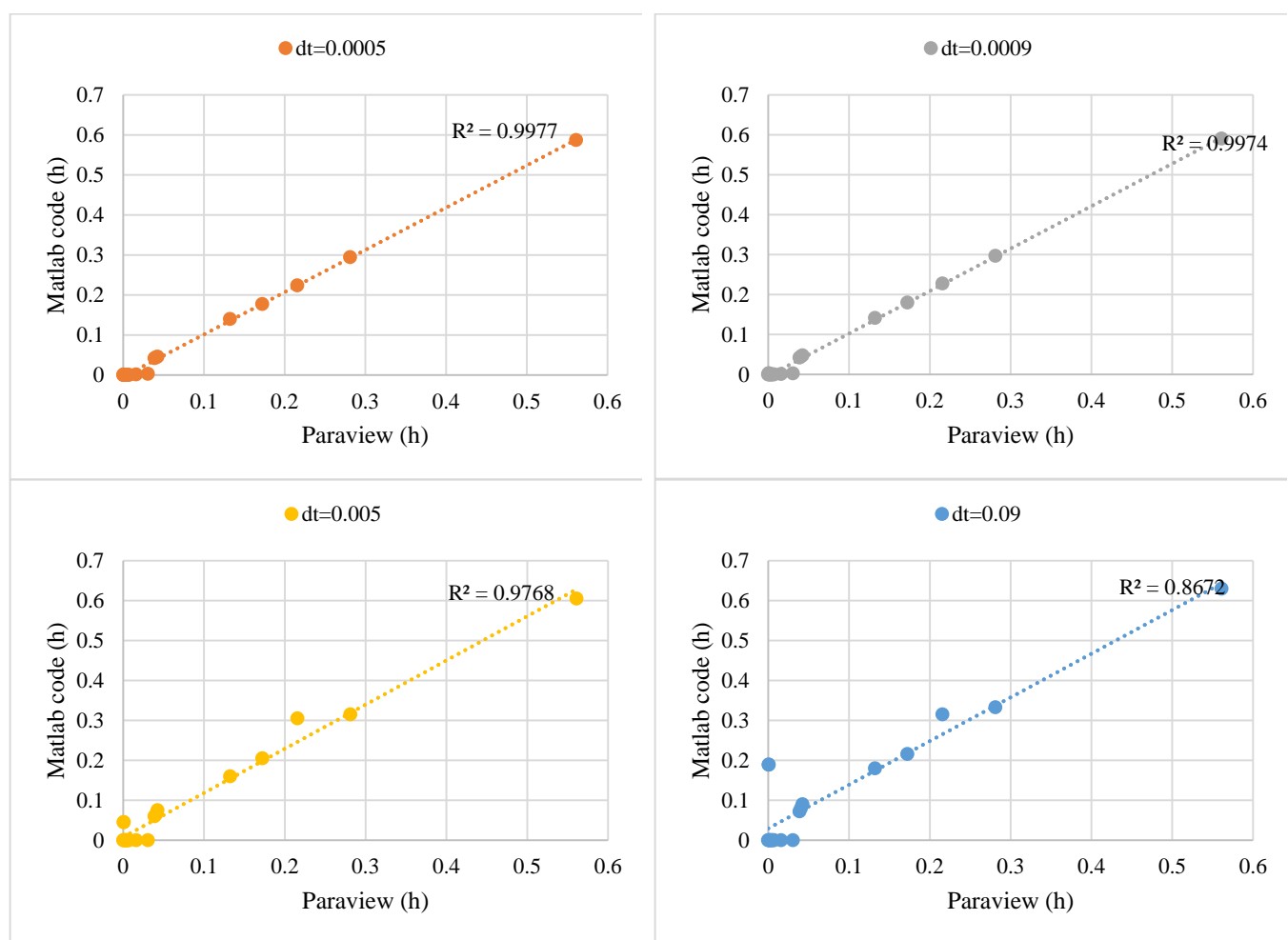

**Figure 8.** Comparison with the residence time obtained from Paraview for different time steps.

### 3.4. Sensitivity Analysis on Spatial and Temporal Discretization

We obtain higher precision in computing residence time by adopting a higher number of control volumes. Velocity components along the y direction are less significant and can be neglected. However, if the number of control volumes is increased, the computational effort/cost increases significantly. By observing the sensitivity of control volume over residence time, it was found that the number of control volumes along x should be in the range of 400 to 500 and z should be 40 to 85. However, if the number of control volumes is increased, the computational cost increases significantly. For example, if 500 and 60 control volumes are considered along x and z directions, respectively, a total of 1,236,316 control volumes are created, which requires higher memory and rendering time. Considering the

computational accuracy and efforts, 450 and 30 control volumes were selected in the x and z directions for the computation. In Figure 9, it can be seen that with an increase in the number of control volumes in the x direction, residence times decrease and reach a stable condition after reaching 200 control volumes, where the change in residence time is negligible with the change in the number of control volumes. On the other hand, residence times increase with the increase in the number of control volumes in the z direction. It also reaches a more stable condition after 25 control volumes, shown in Figure 10.

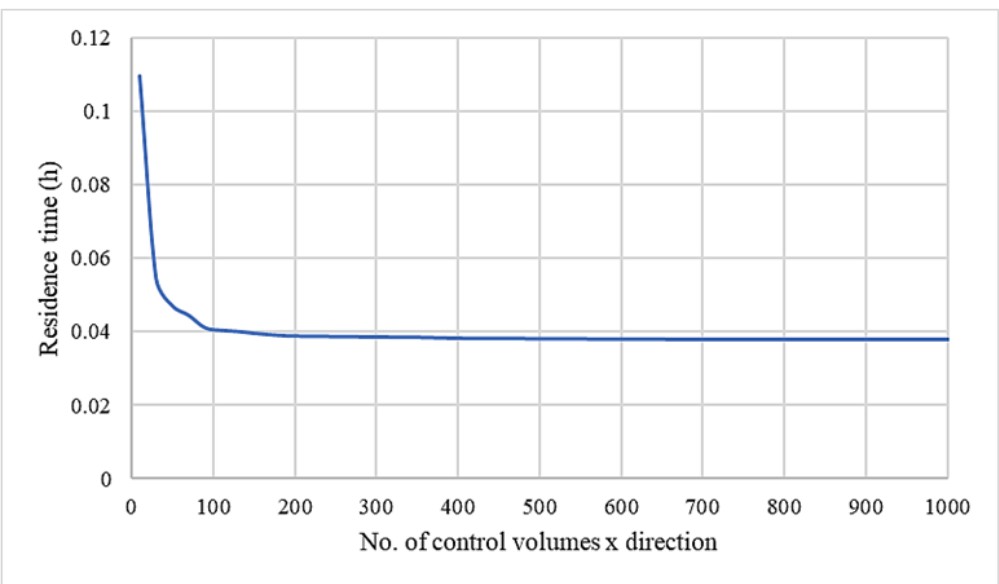

**Figure 9.** Change in residence time with increasing number of control volumes along x direction.

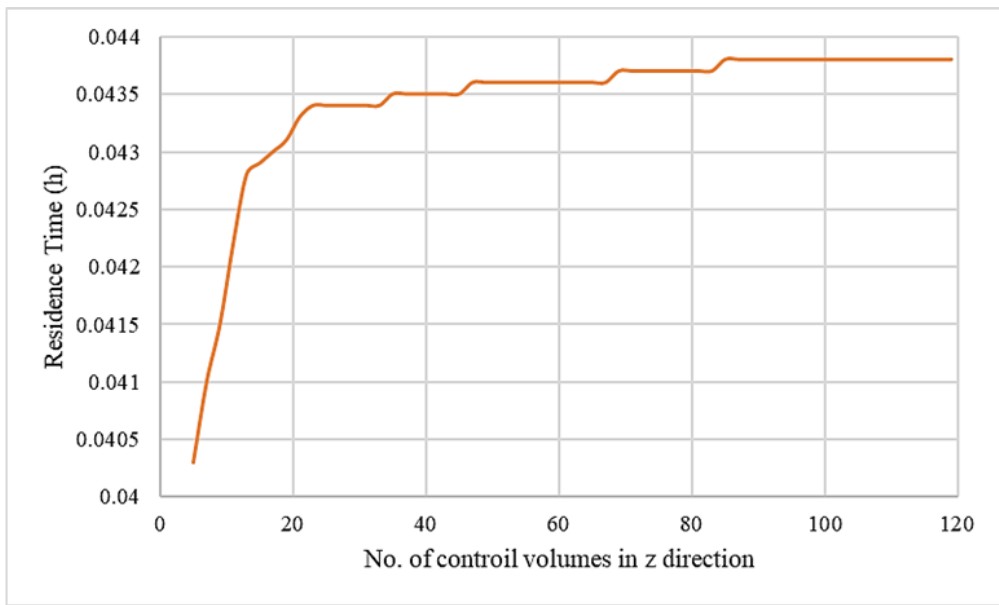

**Figure 10.** Change in residence time with increasing number of control volumes along z direction.

The sensitivity of temporal discretization (computational time step) on the model outcome is analyzed in this study. The computational time step is a vital parameter for performing numerical integration of the streamline trajectories. The accuracy of numerical integration is highly dependent on this parameter. In this study, a fixed value of the computational time step was assumed for calculating streamlines. However, a variety of values is also considered to analyze the impact on residence time. In Figure 11, the

increasing trend of residence times can be seen with increasing computational time steps. As representative seeding locations, x = 0.05, 0.4, 0.8, 0.95, y = 0.5, and z = 0.1 were chosen. In general, residence time shows a slight increase with the increase in the computational time step. The selection of optimal time steps is very crucial for numerical integration accuracy and computational efforts. Considering the smoothness of streamlines and accuracy in the integration method, a constant computational time step of 1 s was considered for the study.

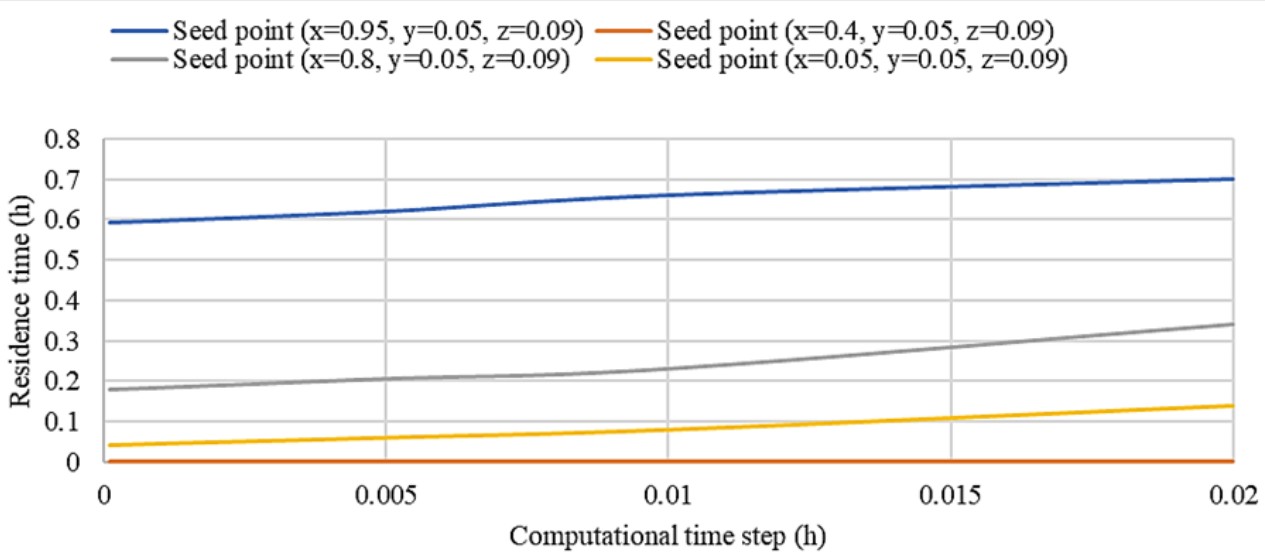

**Figure 11.** Change in residence time (h) with increasing computational time step (h).

### 4. Discussion and Conclusions

Several technical considerations arise when computing particle track. A first concern regards the accuracy of the integration method used. Euler integration has an error of $O(\Delta t^2)$, which means that halving the integration step $\Delta t$ reduces the integration error by a quarter. However, numerical integration has the appalling property that it accumulates errors as the integration time increases since positions along the streamline are computed incrementally. This means, in practice, that the "tails" of long streamlines tend to deviate from their actual correct locations because of having an error of $O(\Delta t^2)$ in each numerical integration of the forward Euler method used in the MATLAB code. The accuracy of integration can be improved by using higher-order methods, e.g., the Runge–Kutta method. This may allow us to increase the time step $\Delta t$ and maintain similar accuracy, which in turn decreases computation effort and rendering time. Many other numerical methods exist for approximating with various trade-offs between accuracy and computational complexity. We refer for further details to the specialized literature.

Setting the integration step $\Delta t$ to small values reduces the errors but at additional costs: the integration takes more time, and the resulting streamline has more sample points, which need more storage and rendering time. Obtaining optimal values for $\Delta t$ is a difficult problem. These depend locally on the data set cell sizes, vector field magnitude, vector field variation, desired particle tracing length and desired computation speed. By "locally," we mean that it is often desirable to adapt $\Delta t$ as the integration proceeds instead of using a constant $\Delta t$ for the complete streamline. Although there is no simple technique for setting $\Delta t$ optimally, there are a few hints in this direction. Using a constant $\Delta t$ is equivalent to a uniform sampling of the integration time dimension. For a vector field of varying magnitude, this obviously produces sample points pi that are spaced irregularly along the streamline or a non-uniform sampling of the spatial dimension. This is often undesirable. Even when using a small $\Delta t$, large vector field values generate large streamline steps that can skip several data set cells, hence under sample the vector field. For a rapidly

varying vector field, this can change the streamline direction dramatically, yielding a misleading visualization.

Mean residence time largely depends on the number of particles seeded. Mean residence time varies substantially when a small amount of particle is seeded. However, with the increase in the number of particles seeded, this variation gets smaller and reaches a stable condition. Seeding locations were chosen to be uniformly distributed in this study project, where in reality, the particle cannot be seeded uniformly over an area. Random choice of seeding location might play an important role in predicting the required number of particles needed for the accurate calculation of streamlines.

**Author Contributions:** M.A.A.M. conceived of the presented idea and developed the algorithm to generate the flow path and residence time in automated fashion, led the investigation of the sensitivity analysis and supervised the findings of the work, carried out the modeling experiment and analysis, and wrote the manuscript. M.M.S.Y. oversaw the investigation, reviewed the algorithm, carried out the modeling experiment and analysis, and oversaw the writing, formatting, and development of the manuscript. All authors have read and agreed to the published version of the manuscript.

**Funding:** This research received no external funding.

**Institutional Review Board Statement:** Not applicable.

**Informed Consent Statement:** Not applicable.

**Data Availability Statement:** Data collected for the study can be made available upon request from the corresponding author.

**Conflicts of Interest:** The authors declare no conflict of interest.

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
