# Peer review of "Automated Particle Tracing & Sensitivity Analysis for Residence Time in a Saturated Subsurface Media"

_liquids, doi:10.3390/liquids2030006_

Round 1
Reviewer 1 Report
Automated Particle Tracing & Sensitivity Analysis for Residence Time in a Saturated Subsurface Media
The authors may be advised to do the following major revisions:
1. Polish the abstract and write it in a precise manner with two or three important findings included over there.
2. The paper has too many subsections reduce them.
3. The text has too long paragraphs. The authors should aim to increase the number of paragraphs to increase readability.
4. Make a new section according to Sensitivity Analysis. Write down the purpose to use the sensitivity in the current study.
5. Write down some detail regarding the used methodology.
6. The authors describe the mathematics involved in the code formulation but do not describe how this was implemented, i.e., how the code is organized and how it was developed. One has no information on whether the authors used an existing code, or whether they have programmed it themselves. If the code was developed by them, did they start from scratch or further developed an existing code?
7. Discuss the behavior of graphs physically.
8. Take care of the formatting of the references.
9. Author should explain the novelty of the current study.
10. Introduction section is very poor. It requires more detail description and linking of the sentences from previous to latest literature.
Author Response
|
Comment |
Response |
Action |
|
##Overall Comments |
||
|
The authors may be advised to do the following major revisions: |
Thank you for your comment. |
N/A |
|
##Comments on the title, abstract, and references |
||
|
1. Polish the abstract and write it in a precise manner with two or three important findings included over there. |
Thank you for the suggestion. The abstract has been updated with concise information and important findings. |
Page:1; Line# 22-25 (Highlighted in bold and yellow) |
|
2. The paper has too many subsections reduce them. |
Thank you for the Comment. Subsections have been reduced. |
Please see the change in the revised manuscript:
Page:6; Line# 190,200,207-216, 260, 278, 307, 330 |
|
3. The text has too long paragraphs. The authors should aim to increase the number of paragraphs to increase readability. |
Thank you for the comment. Number of paragraphs has been increased in the Introduction and methods section. |
Please see the change in the revised manuscript:
Page:2; Line# 60-88 Page:5; Line# 154-164 (Highlighted in bold and yellow) |
|
4. Make a new section according to Sensitivity Analysis. Write down the purpose to use the sensitivity in the current study. |
Thank you for the Comment. A new section with the objectives has been made for the sensitivity analysis. |
Please see the change in the revised manuscript:
Page:6; Line# 199-214 (Highlighted in bold and yellow) |
|
5. Write down some detail regarding the used methodology. |
Thank you for the Comment. Methodology has been detailed with a set of subsections i.e., Automated generation of velocity vector field, particle tracing using MATLAB, Comparison with Paraview and sensitivity analysis on residence time. |
Please see the change in the revised manuscript:
Page:2; Line# 60-76 (Highlighted in bold and yellow) |
|
6. The authors describe the mathematics involved in the code formulation but do not describe how this was implemented, i.e., how the code is organized and how it was developed. One has no information on whether the authors used an existing code, or whether they have programmed it themselves. If the code was developed by them, did they start from scratch or further developed an existing code? |
Thank you for the Comment. MIN3P was used to generate the subsurface flow filed. The output from MIN3P was imported in the MATLAB to generate the streamlines in the subsurface domain. A code was written to generate the streamlines from MIN3P velocity vector field. In addition, the entire process of running MIN3P to create vector files, import them into MATLAB interface and generate streamlines and their corresponding residence times was automated using MATLAB code. |
Please see the change in the revised manuscript:
Page:2; Line# 77-88 (Highlighted in bold and yellow) |
|
7. Discuss the behavior of graphs physically. |
Thank you for the Comment. Behaviors of each graph are described and cited withing the texts. |
Please see the change in the revised manuscript: Page:3; Line# 129 (Highlighted in bold and yellow) Page:5; Line# 181 (Highlighted in bold and yellow) Page:6; Line# 198 (Highlighted in bold and yellow)
Page:7; Line# 227 (Highlighted in bold and yellow) Page:8; Line# 245 (Highlighted in bold and yellow) Page:8; Line# 274 (Highlighted in bold and yellow) Page:9; Line# 280 (Highlighted in bold and yellow) Page:10; Line# 294 (Highlighted in bold and yellow) Page:10; Line# 299 (Highlighted in bold and yellow) Page:11; Line# 318 (Highlighted in bold and yellow) Page:12; Line# 338 (Highlighted in bold and yellow) |
|
8. Take care of the formatting of the references. |
Thank you for the Comment. The manuscript has been updated according to your comment. |
Please see the change in the revised manuscript:
Page:13-14; Line# 400-473 (Highlighted in bold and yellow) |
|
9. Author should explain the novelty of the current study. |
Thank you for the Comment. The novelty of the study has been mentioned the introduction part. |
Please see the change in the revised manuscript:
Page:2; Line# 77-88 (Highlighted in bold and yellow) |
|
10. Introduction section is very poor. It requires more detail description and linking of the sentences from previous to latest literature. |
Thank you for the Comment. Several relevant and recent literatures have been embedded with the scope of the manuscript. |
Please see the change in the revised manuscript:
Page:3; Line# 33-88 (Highlighted in bold and yellow) |

Reviewer 2 Report
This paper focuses on calculating the residence time of particles as an important factor in subsurface media to determine the transport of environmental toxins, the sediment metabolic rates in fluvial ecology as well as hydrological water budget. A rectangular model domain is set up considering a fully saturated porous media under steady state condition in MIN3P. A constant head boundaries condition were parameterized by a linear drop in hydraulic head in upstream and downstream of the model domain. Streamlines and residence times of the particles are computed with a variety of seeding locations covering the whole model surface. Sensitivity analysis for residence time is performed over the varying spatial discretization and computational time steps. Moreover, a comparative study of the outcomes with Paraview is undertaken to validate the automated model. The results of the automated process show that the computed residence times are highly dependent on the accuracy of the integration method, the value of the computational time step, ∆t, spatial discretization, stopping criterion for the integration process of streamlines, the choice of the location, and the number of seed points. The automated process can be highly beneficial in obtaining insights into subsurface flow dynamics with high variability in the model setup instead of laboratory-based experimentation in a computationally efficient manner. The paper is well written and also well organized. It deserves publication in Liquids journal after minor revision. Should the sign in the left hand side of Eq. 3 be +ve, by substituting Eq. (2) into Eq. (1)? If so, it may be required to resolve the model equations.
Author Response
|
Comment |
Response |
Action |
|
Recommendation: |
||
|
Minor revision. |
|
|
|
Comments: |
||
|
This paper focuses on calculating the residence time of particles as an important factor in subsurface media to determine the transport of environmental toxins, the sediment metabolic rates in fluvial ecology as well as hydrological water budget. A rectangular model domain is set up considering a fully saturated porous media under steady state condition in MIN3P. A constant head boundaries condition were parameterized by a linear drop in hydraulic head in upstream and downstream of the model domain. Streamlines and residence times of the particles are computed with a variety of seeding locations covering the whole model surface. Sensitivity analysis for residence time is performed over the varying spatial discretization and computational time steps. Moreover, a comparative study of the outcomes with Paraview is undertaken to validate the automated model. The results of the automated process show that the computed residence times are highly dependent on the accuracy of the integration method, the value of the computational time step, ∆t, spatial discretization, stopping criterion for the integration process of streamlines, the choice of the location, and the number of seed points. The automated process can be highly beneficial in obtaining insights into subsurface flow dynamics with high variability in the model setup instead of laboratory-based experimentation in a computationally efficient manner. The paper is well written and also well organized. It deserves publication in Liquids journal after minor revision. |
Thank you for your valuable suggestions. |
N/A |
|
1. Should the sign in the left hand side of Eq. 3 be +ve, by substituting Eq. (2) into Eq. (1)? If so, it may be required to resolve the model equations. |
Thank you for your suggestion.
|
Please see the change in the revised manuscript: Page:2; Line# 89 (Highlighted in bold and yellow) |

Reviewer 3 Report
A review of the “Automated Particle Tracing & Sensitivity Analysis for Residence Time in a Saturated Subsurface Media”
The topic addressed in this paper is indeed interesting and should be constantly studied. The manuscript is well written and the English is good despite some typos and grammatical errors. Introduction provides relevant information to support the importance of the topic and the purpose of the research is well defined. Research methodology is well discussed with relevant information. Details are provided on the materials and methods. The results are analyzed coherently, in a logical sequence of ideas, in accordance with the investigated parameters. However, the authors are advised to address and make necessary corrections regarding the issues pointed at below:
Line#18, 193, 273, 291 grammatical errors such as subject-verb disagreement
Line#37 unnecessary parenthesis
Line#57 a brief introduction of the Richard’s equation is needed
Line# 66 proper use of indefinite article
Line#70-72; 106;124; 138, 269, 272 proper use of punctuation marks (e.g., “respectively” should set off with a comma)
Line# 74 duplicate auxiliary verb
Line#94 space between value and unit
Line#103 briefly include why Dirichlet was used not Neumann
Line#120 (and throughout the manuscript) ensure a space both before and after = sign
Line#130 Min3P not Min3p
Line#134, 169, 254, 308 No need to capitalize “Forward”
Line#139 Figure 9? Is the figure number correct here? Otherwise, why it is placed later while some discussion around it begins here?
Line#152-153 object in the sentence is missing
Line#159 check for missed spacing
Line# 161, 285 why is “Computational” capitalized?
Line#170 (and throughout the manuscript) “particle’s current position” not “particles current position”
Line#214, 230 inconsistent significant figures
Line#222-224; 226-227; 253-255 unclear sentence structure
Line#241 “R2” not “R2”
Line#256 figure 22?
Line# 264 why is “Discretization” capitalized?
Line#285 remove “:”
Nevertheless, the scope of the study matches the scientific scope of the journal. The results are well described together with an acceptable scientific discussion and conclusion. Therefore, the manuscript is recommended for publication after the issues mentioned above are taken care of.
Author Response
|
Comment |
Response |
Action |
|
##Overall Comments |
||
|
The topic addressed in this paper is indeed interesting and should be constantly studied. The manuscript is well written and the English is good despite some typos and grammatical errors. Introduction provides relevant information to support the importance of the topic and the purpose of the research is well defined. Research methodology is well discussed with relevant information. Details are provided on the materials and methods. The results are analyzed coherently, in a logical sequence of ideas, in accordance with the investigated parameters. However, the authors are advised to address and make necessary corrections regarding the issues pointed at below: |
Thank you for your comment. |
N/A |
|
##Comments on the title, abstract, and references |
||
|
1. Line#18, 193, 273, 291 grammatical errors such as subject-verb disagreement |
Thank you for the suggestion. We have changed the manuscript according to the comments. |
Page:1; Line# 18-19 (Highlighted in bold and yellow) Page:7; Line# 231-233 (Highlighted in bold and yellow) Page:7; Line# 316-318 (Highlighted in bold and yellow) Page:7; Line# 337-338 |
|
2. Line#37 unnecessary parenthesis
|
Thank you for your observation. We have changed the manuscript according to the comments. |
Please see the change in the revised manuscript:
Page:1; Line# 37 (Highlighted in bold and yellow) |
|
3. Line#57 a brief introduction of the Richard’s equation is needed
|
Thank you for the Suggestion. We have changed the manuscript according to the comments. |
Please see the change in the revised manuscript:
Page:3; Line# 95-99 (Highlighted in bold and yellow) |
|
4. Line# 66 proper use of indefinite article
|
Thank you for the Comment. We have changed the manuscript according to the comments. |
Please see the change in the revised manuscript:
Page:3; Line# 86-88 (Highlighted in bold and yellow) |
|
5. Line#70-72; 106;124; 138, 269, 272 proper use of punctuation marks (e.g., “respectively” should set off with a comma)
|
Thank you for the Comment. We have changed the manuscript according to the comments. |
Please see the change in the revised manuscript:
Page:3; Line# 93, 120,124,135 (Highlighted in bold and yellow) Page:4; Line# 152 (Highlighted in bold and yellow) Page:5; Line# 176 (Highlighted in bold and yellow) Page:11; Line# 315 (Highlighted in bold and yellow) |
|
6. Line# 74 duplicate auxiliary verb
|
Thank you for the Comment. We have changed the manuscript according to the comments. |
Please see the change in the revised manuscript:
Page:3; Line# 94 (Highlighted in bold and yellow) |
|
7. Line#94 space between value and unit
|
Thank you for the Comment. We have changed the manuscript according to the comments. |
Please see the change in the revised manuscript:
Page:3; Line# 120 (Highlighted in bold and yellow) |
|
8. Line#103 briefly include why Dirichlet was used not Neumann
|
Thank you for the Comment. We have changed the manuscript according to the comments. |
Please see the change in the revised manuscript:
Page:3; Line# 130-132 (Highlighted in bold and yellow) |
|
9. Line#120(and throughout the manuscript) ensure a space both before and after = sign
|
Thank you for the Comment. We have changed the entire manuscript according to the comments. |
Please see the change in the revised manuscript:
Page:1; Line# 23 Page:3; Line# 129 Page:4; Line# 137, 148-149 Page:5; Line# 160,175,180, Page:7; Line# 226,230-232, 237, 243-244 Page:8; Line# 249, 252-253, 258, 263-264, 269 Page:9; Line# 287 Page:10; Line# 293-294 Page:12; Line# 340
(Highlighted in bold and yellow) |
|
10. Line#130 Min3P not Min3p
|
Thank you for the Comment. We have modified the manuscript. |
Please see the change in the revised manuscript:
Page:5; Line# 168 (Highlighted in bold and yellow) |
|
11. Line#134, 169, 254, 308 No need to capitalize “Forward”
|
Thank you for the Comment. We have modified the manuscript. |
Please see the change in the revised manuscript:
Page:5; Line# 172 Page:6; Line# 208 Page:10; Line# 297 Page:12; Line# 356 (Highlighted in bold and yellow) |
|
12. Line#139 Figure 9? Is the figure number correct here? Otherwise, why it is placed later while some discussion around it begins here?
|
Thank you for the Comment. We have removed the term. It was a typo. |
Please see the change in the revised manuscript:
Page:5; Line# 177 (Highlighted in bold and yellow) |
|
13. Line#152-153 object in the sentence is missing
|
Thank you for the Comment. We have modified the manuscript. |
Please see the change in the revised manuscript:
Page:5; Line# 185 (Highlighted in bold and yellow) |
|
14. Line#159 check for missed spacing
|
Thank you for the Comment. We have modified the manuscript. |
Please see the change in the revised manuscript:
Page:6; Line# 192 (Highlighted in bold and yellow) |
|
15. Line# 161, 285 why is “Computational” capitalized?
|
Thank you for the Comment. Capitalized “Computational” has been changed to regular format. |
Please see the change in the revised manuscript:
Page:6; Line# 194 Page:11; Line# 313
(Highlighted in bold and yellow) |
|
16. Line#170 (and throughout the manuscript) “particle’s current position” not “particles current position”
|
Thank you for the Comment. We have modified the manuscript. |
Please see the change in the revised manuscript:
Page:5; Line# 170 Page:5; Line# 175 Page:6; Line# 208 Page:6; Line# 212 (Highlighted in bold and yellow) |
|
17. Line#214, 230 inconsistent significant figures
|
Thank you for the Comment. We have modified the manuscript. |
Please see the change in the revised manuscript:
Page:8; Line# 253, 256 Page:8; Line# 271
(Highlighted in bold and yellow) |
|
18. Line#222-224; 226-227; 253-255 unclear sentence structure
|
Thank you for the Comment. We have revised the sentences. |
Please see the change in the revised manuscript Page:8; Line# 261-267, 271-274
|
|
19. Line#241 “R2” not “R2”
|
Thank you for the Comment. We have corrected to R2 |
Please see the change in the revised manuscript:
Page:9; Line# 283 (Highlighted in bold and yellow) |
|
20. Line#256 figure 22?
|
Thank you for the Comment. We have modified the manuscript. |
Please see the change in the revised manuscript:
Page:10; Line# 299 (Highlighted in bold and yellow) |
|
21. Line# 264 why is “Discretization” capitalized?
|
Thank you for the Comment. We removed capitalized “Discretization”. |
Please see the change in the revised manuscript:
Page:10; Line# 306 |
|
22. Line#285 remove “:” |
Thank you for the Comment. We have removed ‘:’. |
Please see the change in the revised manuscript:
Page:11; Line# 323 |
|
23. Nevertheless, the scope of the study matches the scientific scope of the journal. The results are well described together with an acceptable scientific discussion and conclusion. Therefore, the manuscript is recommended for publication after the issues mentioned above are taken care of. |
Thank you for the Comment. |
N/A |
